# LLM CASCADE WITH MULTI-OBJECTIVE OPTIMAL CONSIDERATION

## ABSTRACT

Large Language Models (LLMs) have demonstrated exceptional capabilities in understanding and generating natural language. However, their high deployment costs often pose a barrier to practical applications, especially. Cascading local and server models offers a promising solution to this challenge. While existing studies on LLM cascades have primarily focused on the performance-cost trade-off, real-world scenarios often involve more complex requirements. This paper introduces a novel LLM Cascade strategy with Multi-Objective Optimization, enabling LLM cascades to consider additional objectives (e.g., privacy) and better align with the specific demands of real-world applications while maintaining their original cascading abilities. Extensive experiments on three benchmarks validate the effectiveness and superiority of our approach.

## 1 INTRODUCTION

As Large Language Models (LLMs) continue to evolve rapidly (Touvron et al., 2023; Achiam et al., 2023; Reid et al., 2024), they are increasingly being integrated into real-world applications, enhancing the intelligence of a wide range of systems. At the same time, mobile devices have become indispensable in everyday life. The emergence of on-device intelligence—such as Apple Intelligence (Gunter et al., 2024) and Gemini Live (Reid et al., 2024)—which embeds LLMs directly into devices for more personalized and intelligent user interactions, is gaining traction but remains relatively underexplored (Xu et al., 2024). A major challenge in this area is the hardware limitations of mobile devices, including constraints on compute power, battery life, and storage capacity. As a result, only smaller LLMs, such as Gemma-2B (Team et al., 2024), can be deployed on these devices, leading to trade-offs in performance compared to larger, more powerful models like Gemini. This raises a critical question for the research community: how can we optimize on-device intelligence given these size constraints? The LLM cascade method presents a solution for this challenge.

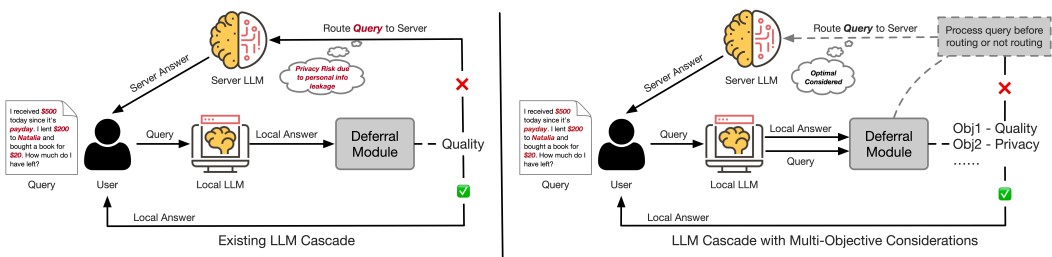

Figure 1: On the right is the existing LLM cascade, where the deferral module makes decisions solely based on the quality of the local answer, potentially leading to privacy leakage. On the left is our proposed LLM cascade with multi-objective considerations, where deferral decisions are more aligned with the needs of real-world applications.

In an LLM cascade system, a query is usually first processed by a smaller, weaker local LLM and is only escalated to a larger, stronger server LLM if the local model's output is deemed insufficient by a deferral module, as shown in Figure 1. This paradigm has garnered significant attention recently

(Chen et al., 2023a; Gupta et al., 2024; Yue et al., 2023; Wang et al., 2024). As larger LLMs are often substantially more expensive than their smaller counterparts (e.g., Gemini-1.5 Pro (Reid et al., 2024) costs up to 35 times more than Gemini-Flash[1]), most existing LLM cascade works focused on the exploration of optimal trade-offs between cost and performance. However, real-world applications can be more complicated and requires the cascade system to make deferral decisions beyond just performance-cost consideration. For instance, privacy concerns may arise if personal data is routed to the server LLM where decisions are made based solely on the local answer's quality, as illustrated in Figure 1. Unfortunately, few studies have explored the LLM cascade with multi-objective consideration. To address this, we propose to endorse multi-objective optimal considerations into the decision making by the LLM cascade system where the deferral module may hesitated to route the user query not only considering the local answer's quality but also with other considerations (e.g., privacy) as depicted in Figure 1.

One key focus of LLM cascade research is the design of deferral criteria, which determine whether a query needs to be routed to the server model. Ideally, the deferral criteria should identify queries that the local LLM is unlikely to handle effectively, sending them to the server to significantly improve performance while keeping costs manageable. Conversely, sending queries that the local LLM can address with high quality to the server can result in unnecessary costs. Intuitively, model confidence could serve as a good indicator, with queries routed to the server when the local model is not confident with its response. For instance, Zhu et al. (2024) explored a self-critique strategy to leverage the local model's intelligence to produce a confidence level in terms of the local answer and make decisions based on the confidence level. However, Jitkrittum et al. (2024) noticed the weakness of confidence-based deferral rule in cases where distribution shifts occur between the training and test datasets. Logit-based methods step further by using the generated token logits of the local answer as features to make deferral decisions. For example, Gupta et al. (2024) found the length bias and token uncertainty problems in cascading by relying on the mean logits and proposed to leverage quantile logits as features to mitigate this problem. Additionally, Wang et al. (2024) introduced cascade-aware training, which incorporates both the local and server LLM's logits into the loss function during local model training, helping the local LLM become more aware of which queries should be deferred to the server. Unfortunately, none of these works explored deferral decision making with respects to other objectives such as privacy. To address this gap, we propose incorporating multi-objective optimization into the LLM cascade system. The key is to enable the local LLM to better understand multi-objective deferral logic, rather than focusing solely on the cost-performance trade-off. Intuitively, we can utilize the in-context learning abilities of the local model by designing appropriate instructional prompts to help it understand the cascade logic with multi-objective considerations (Sordoni et al., 2024; Hartmann et al., 2024). However, this approach is limited by the size and corresponding in-context learning capacity of the local LLM. Another option is training the local LLM to incorporate multi-objective considerations. Instruction tuning has proven highly effective at improving LLM performance across specific tasks, as well as enhancing its ability to follow instructions (Zhao et al., 2024; Chen et al., 2024; Ma et al., 2024), aligning well with our goal of embedding cascade logic into the local model. Moreover, incorporating the more powerful server LLM's capabilities into the customized loss function during local LLM training penalizes the local model for producing high logits associated with poor-quality outputs(Wang et al., 2024). In tandem, we explore both training-based methods (i.e., instruction tuning, loss tuning) and training-free approaches (i.e., prompt engineering) to enable the local LLM to account for multi-objective considerations when deciding whether to invoke the server model. The contributions of this study are three-fold:

• We extend the current focus of LLM cascading beyond the traditional cost-performance trade-off to include multi-objective considerations, better aligning with the needs of real-world applications.

• We explore both training and training-free methods to enable local LLMs to comprehend complex cascade logic with multi-objective considerations.

• Extensive experiments on three benchmarks have validated the necessity and superiority of incorporating multi-objective considerations into LLM cascading, rather than relying solely on cost-performance trade-offs[2].

---

[1]https://ai.google.dev/pricing

[2]To encourage further explorations by the community, we will open-source our implementations (a copy is attached with this submissions).

## 2 METHODOLOGY

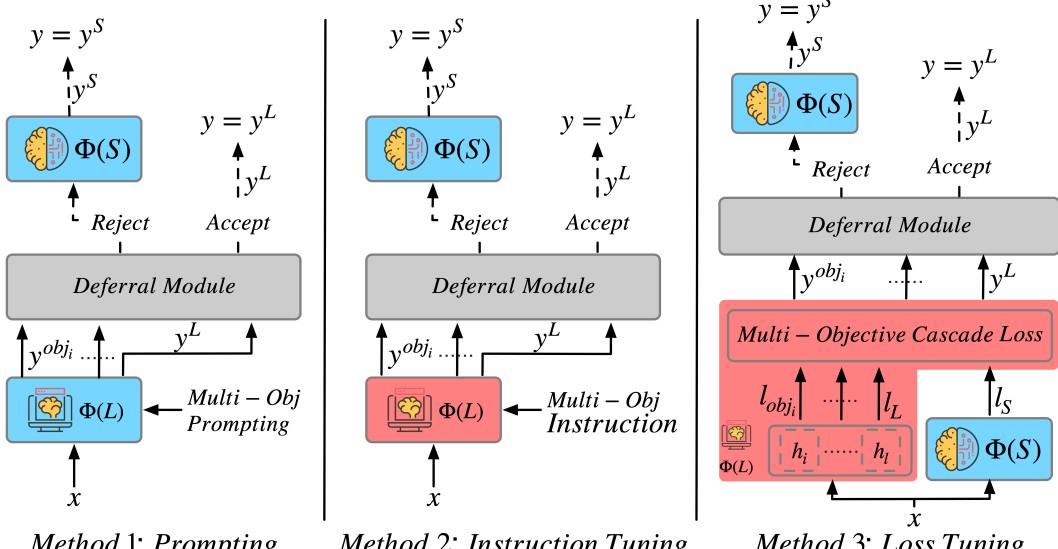

Figure 2: Overview of the proposed methods: $\Phi(L)$ and $\Phi(S)$ represent the local model and server model, respectively. The red box indicates trainable, while the blue box represents frozen. $\Phi(L)$ is tasked with generating responses $y^L$ and $y^{obj_i}$ for both the query $x$ and the multi-objective considerations $obj_i$. For loss tuning, the generation tasks are handled by different heads $h_i$, and a combined cascade loss is utilized for tuning.

### 2.1 PRELIMINARY FORMULATION

Before proceeding, we will first present the preliminary concepts and formulations. Given a local LLM $\Phi(L)$ (smaller and weaker) and a server LLM $\Phi(S)$ (larger and stronger), when a user sends a query $x$ to $\Phi(L)$, the local model generates an initial answer $y^L$. A deferral module $D(\cdot)$ then determines whether it is necessary to route the query $x$ to $\Phi(S)$. If $D(\cdot)$ accepts $y^L$, it becomes the final answer $y$ returned to the user. If rejected, the query $x$ is routed to $\Phi(S)$, and the server-generated answer $y^S$ serves as the final response $y$. Our objective in this study is to enable $\Phi(L)$ to be aware of multi-objective considerations $[obj_1, ..., obj_i]$ while generating $y^L$. The responses $[y^{obj_1}, ..., y^{obj_i}]$ corresponding to these considerations, along with $y^L$, can be utilized in $D([y^{obj_1}, ..., y^{obj_i}, y^L])$ to inform decision-making. In this study, we primarily focus on two objectives: privacy and quality. In the following sections, we will illustrate how to incorporate multi-objective considerations into both training methods (instruction tuning and loss tuning) and training-free methods (prompting).

### 2.2 MULTI-OBJECTIVE PROMPTING

Ideally, the $\Phi(L)$ can be taught multi-objective optimal cascade logic based on its own natural language understanding ability. Efforts have been made to enable the $\Phi(L)$ being aware of the confidence of generated responses via self-critique(Zhu et al., 2024), step-by-step prompting(Zhang & Gao, 2023) etc. We step further on the previous works and include the privacy concern (Hartmann et al., 2024) into prompt design. Specifically, we formulate an instructional prompt[3] which integrates query $x$ and objective considerations (i.e., privacy consideration $obj_p$) to the $\Phi(L)$ to obtain response $[y^{obj_p}, y^L]$, and these response will further be sent to the $D(\cdot)$ where deferral decisions will be made. Further, we follow Deng et al. (2024)'s work and perform few-shot prompting to better activate the $\Phi(L)$'s in-context learning ability. However, with limited size, the $\Phi$ is inadequate[4] to understand the

---

[3]The prompts used can be seen in the appendix A

[4]Please refer to the appendix B for better understanding over the local llm's weakness.

multi-objective optimal cascade logic relying its own ability and the complicated logic may further hurt its ability to answer user's query and thus training is needed.

## 2.3 Multi-Objective Instruction Tuning

Previous studies have demonstrated the effectiveness of instruction tuning in enhancing downstream task performance and improving comprehension of given instructions (Zhu et al., 2024; Zhao et al., 2024; Ma et al., 2024; Li et al., 2023). This ability to understand instructions aligns well with our objective of grasping the deferral logic. Furthermore, the improvements in task performance help mitigate any negative impacts on generating $y^L$ that may arise from producing $y^{obj_i}$ during prompting. Similar to the prompting method, we utilize an instructional prompt that combines a step-by-step instruction with the user query $x$ as input. The labeled text $\hat{y}$ corresponding to $x$, along with the labeled responses $\hat{y}^{obj_i}$ for the multi-objective considerations, serve as outputs for fine-tuning the model $\Phi(L)$. The responses generated by the tuned model will then be utilized by the deferral module $D(\cdot)$ to determine whether routing to the server model $\Phi(S)$ is necessary.

## 2.4 Multi-Objective Loss Tuning

Stepping further over the methods that rely on the local model's intricate understanding ability, recent works have pointed out the superiority of distilling the server llm's ability on downstream tasks into the loss function for tuning the local model(Wang et al., 2024). Intuitively, our assumption is that the server llm is larger and more powerful(Hartmann et al., 2024) in terms of down-stream tasks, and thus the discrepancy between the generations of $\Phi(L)$ and $\Phi(S)$ can somehow be used for $\Phi(L)$ to indicate the confidence level. The larger the discrepancy is, the lower confidence level should the $\Phi(L)$ have. However, to enable $\Phi(L)$ being aware of multi-objective considerations, simply including the distillation loss from $\Phi(S)$ is inadequate. To this end, we decompose the overall task into several sub-tasks and use different heads to handle the different sub-tasks. Namely, given the multi-objective considerations $[obj_1, ..., obj_i]$ and the query $x$, we leverage multiple llm heads $[h_1, ..., h_i, h_L]$ to handle different considerations and the query. Each head will produce a loss and a distillation loss from $\Phi(S)$ will be optionally added. These losses will then be sent to a weighted-sum function to produce a multi-objective cascade loss for tuning $\Phi(L)$:

$$l = \sum_i^n w_i \cdot l_{obj_i} + w_L \cdot l_L + \alpha(t) \cdot w_S \cdot l_S$$

$$\sum_i^n w_i^n + w_L + w_S = 1, \alpha(t) = H(logit_{y^L}, t) \tag{1}$$

where $w_i$ denotes the weight for the loss associated with generating response $y^{obj_i}$ for the objective $obj_i$, $w_L$ is the weight for the loss of generating response $y^L$ for $x$ from $\Phi(L)$ and $w_S$ is the weight for the loss of generating response $y^S$ for $x$ from $\Phi(S)$. $n$ is the number of objectives that need to be considered. $\alpha$ is the factor for controlling if the knowledge from the server LLM $\Phi(S)$ is used depending on a logit threshold $t$. $H(\cdot, t)$ is a modified Heaviside Step function which returns 0 if $\cdot > t$ else returns 1. In the context of identifying privacy concern, the loss function we utilized for tuning $\Phi(L)$ is:

$$l = - w_p \cdot (\hat{y}^p \cdot log(p_L(y^p|x)) + (1 - \hat{y}^p) \cdot log(1 - p_L(y^p|x))) +$$

$$w_L \cdot log(p_L(y^L|x)) + \alpha(t) \cdot w_S \cdot log(p_S(y^S|x)) \tag{2}$$

where $y^p$, $\hat{y}^p$ are the predicted, golden binary predictions for privacy, respectively. Other terms remain the same as in formula 1. By incorporating multi-objective considerations into the loss function for tuning $\Phi(L)$, the model will generate answers with better awareness of these considerations. The corresponding logits of the generated answers by tuned $\Phi(L)$ can then be utilized by the deferral module to inform decision-making.

## 2.5 Deferral Module

All the three methods are studying how to enable the local LLM to be aware of multi-objective considerations while generating the response to the query. And such considerations are presented

as the logit distributions of the generated response, for example, higher logit may indicated higher performance and less privacy concern. Deferral module plays a pivotal role in the LLM cascade since it decides which query to send out to the server llm based on the logits. Following previous successes on using different logit (e.g., mean, quantile) of the generated response as the reference to decide if there is a need to route the query to the server LLM(Wang et al., 2024; Jitkrittum et al., 2024; Gupta et al., 2024), we also utilize the logit of generated response as indicators to make the routing decisions. Specifically, given a threshold $t \in (0, 1)$, if the logit of the generated response exceed $t$ then it means the local LLM is confident with its response and no need to route, otherwise route the query $x$ to the server LLM $\Phi(S)$.

## 3 EXPERIMENTAL SETTINGS

### 3.1 DATASETS

To validate the effectiveness of including multi-objective considerations into LLM cascade, we opt for three benchmarks to test our methods as below, more statistics can be seen in appendix C.2.

**GSM8K**(Cobbe et al., 2021) is a graduate student mathematical dataset consisting of mathematical questions and corresponding solutions, of which some questions contain personal information for privacy study(Hartmann et al., 2024).

**MedQSum**(Zekaoui et al., 2023) is a medical related dataset with a focus on summarizing the customer health question. The dataset contains customer health questions and corresponding summaries which contains personal healthcare information.

**WMT22**(Kocmi et al., 2022) is a sequence-to-sequence translation dataset consisting of source language sentences and corresponding target language sentences.

### 3.2 TASKS & METRICS

| Dataset | Task Type | Privacy? | Measurement |
|---------|-----------|----------|-------------|
| GSM8K | Question Answering | ✓ | Accuracy, Privacy Leakage |
| MedQSum | Summarization | ✓ | ROUGE, Privacy Leakage |
| WMT22 | Translation | ✗ | ROUGE |

Table 1: Details of tasks and measurements.

We evaluate our proposed LLM cascade with multi-objective optimal considerations on three commonly used tasks: Question Answering, Summarization, and Translation, as indicated in Table 1. For datasets involving privacy concerns, we also incorporate the metric of privacy leakage (Hartmann et al., 2024), which calculates the average number of privacy tokens leaked when sending queries to the server LLM (Check more details in appendix C.2). This approach demonstrates the necessity and effectiveness of considering multi-objective factors in the LLM cascade.

### 3.3 BASE MODELS & IMPLEMENTATION DETAILS

For implementation details, we leverage the Transformers(Wolf et al., 2020) as the base code and conduct extensive experiments with the Gemma models(Team et al., 2024): **Gemma-2B** as the local LLM, **Gemma-7B** as the server LLM. Notably, the server LLM is fine-tuned on all datasets to reach reasonably great performance, of which the server LLM's ability on GSM8K, MedQSum and WMT22 are 52.85%, 61.22% and 36.51%, respectively. We use the AdamW optimizer(Loshchilov & Hutter, 2018; Paszke et al., 2017) with a learning rate of 5e-4 and also a linear warm-up scheduler initialized with 10% of the total training steps as warm-up steps and a weight decay of 1e-4 to avoid over-fitting for all the experiments. The batch size per device is set to 8. All the experiments are conducted on two computation nodes configured with eight 80G H100 GPUs.

# 4 EXPERIMENTAL RESULTS

## 4.1 CASCADE STUDY

| Dataset | Metric | % | Prompt Engineering 0-shot | few-shot | Instruction Tuning | Loss Tuning |
|---------|--------|---|------|------|------|------|
| GSM8K | CR | | 100 | 100 | 100 | **81.2** |
| | SCR | | 28.13 | 28.13 | 28.13 | **31.75** |
| | | $\Phi(L)$ | 14.94 | 11.83 | 26.08 | 26.91 |
| | Acc | $\Phi(L) + \Phi(S)$ | 52.85 | 52.85 | 52.85 | **55.92** |
| | | vs $\Phi(S)$ | - | - | - | ↑**3.07** |
| MedQSum | CR | | 99.3 | 96.2 | **94.8** | 97.3 |
| | SCR | | 25.98 | 26.09 | 26.89 | **26.92** |
| | | $\Phi(L)$ | 21.69 | 28.55 | 34.61 | 36.77 |
| | R-S | $\Phi(L) + \Phi(S)$ | 61.81 | 61.97 | 62.18 | **62.95** |
| | | vs $\Phi(S)$ | ↑0.59 | ↑0.75 | ↑0.96 | ↑**1.73** |
| WMT22 | CR | | 100 | 90.9 | 94.7 | **80.6** |
| | | $\Phi(L)$ | 6.22 | 8.36 | 11.49 | 14.58 |
| | R-S | $\Phi(L) + \Phi(S)$ | 36.51 | 37.39 | 39.04 | **39.69** |
| | | vs $\Phi(S)$ | - | ↑0.88 | ↑2.53 | ↑**3.18** |

Table 2: Table 2 presents the best cascade performance of $\Phi(L)$ across three benchmarks. CR denotes the call rate, indicating the proportion of queries sent to the server. SCR represents the safe call rate, reflecting the number of queries that are safe (i.e., those sent to the server that do not contain privacy information) among the total sent queries. Acc refers to accuracy, while R-S indicates the ROUGE-Sum score. The symbol ↑ signifies an improvement compared to $\Phi(S)$.

**Cascade Performance** As shown in Table 2, the cascade approach significantly enhances the performance of the local model $\Phi(L)$, even surpassing the server model $\Phi(S)$. For instance, by routing 81.2% of queries to the server, the loss-tuned $\Phi(L)$ achieves a 55.92% accuracy on the GSM8K dataset, reflecting a 3.07% improvement over $\Phi(S)$. On the MedQSum dataset, improvements in rouge-sum scores of 0.59%, 0.75%, 0.96%, and 1.73% are observed for 0-shot prompting, few-shot prompting, instruction tuning, and loss tuning, respectively, with routing rates of 99.3%, 96.2%, 94.8%, and 97.3%. A similar pattern is noted on the WMT22 dataset, further validating the advantages of LLM cascade for the local model $\Phi(L)$. However, the cost of cascading remains a critical concern in real-world applications. The goal of the cascade is to enhance the local model's performance while maintaining a reasonable server call rate. We observe that training-based methods, such as instruction tuning and loss tuning, yield larger performance gains at lower call rates, indicating the necessity of training the local model to optimize cost-performance trade-offs. In contrast, the performance of training-free methods (e.g., prompt engineering) heavily depends on the server model $\Phi(S)$, rather than the cascade itself. For example, on the GSM8K dataset, the best performance of training-free methods coincides with sending all queries to the server, a pattern is also seen on the WMT22 dataset. This suggests that the local model struggles to identify which queries should be routed to the server. Furthermore, training methods demonstrate a more favorable "safe call" rate compared to training-free methods, highlighting the local model's inability to incorporate multi-objective considerations during cascading. This underscores the need to include multi-objective optimization strategies in LLM cascading.

**Performance vs Cost** To further understand how the call rate impacts on the local LLM's performance, we set different thresholds $t$ ranging from 0 to 1 with a step of 0.05 to see the performance trends on three datasets. As can be observed in Figure 3, both 0-shot prompting and few-shot prompting exhibit a roughly linear performance improvement as the call rate increases on the GSM8K and MedQSum datasets, suggesting that the prompting methods tend to route queries randomly. However, on the WMT22 dataset, the performance curve for the prompting methods suggests that the local LLM struggles to grasp cascade logic when considering other objectives. In contrast, training methods, especially loss tuning, display a performance increase curve as the number of calls rises, with specific inflection points indicating the optimal trade-off between performance and cost. For instance, when constrained to a 50% call rate, loss tuning demonstrates the best performance, even matching the capabilities of the server LLM, which is quite promising. These

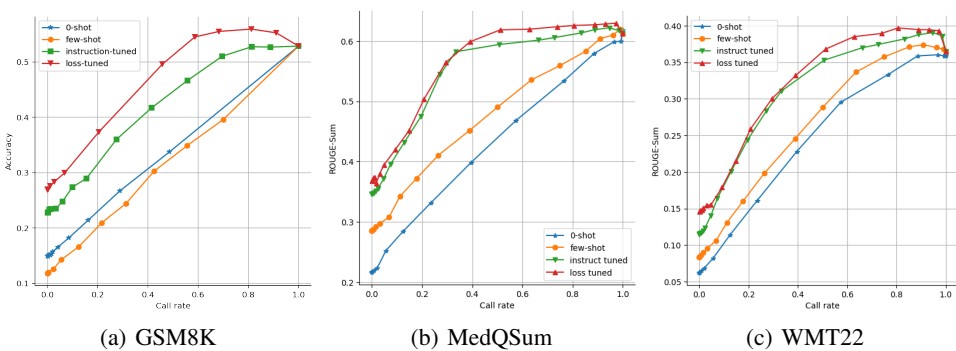

(a) GSM8K        (b) MedQSum        (c) WMT22

Figure 3: Curves depicting cascade performance versus call rate for different methods across all three datasets: (a) GSM8K, (b) MedQSum, and (c) WMT22.

observations reinforce the necessity for training the local model to effectively understand cascade logic, particularly when incorporating multi-objective considerations.

## 4.2 PRIVACY STUDY

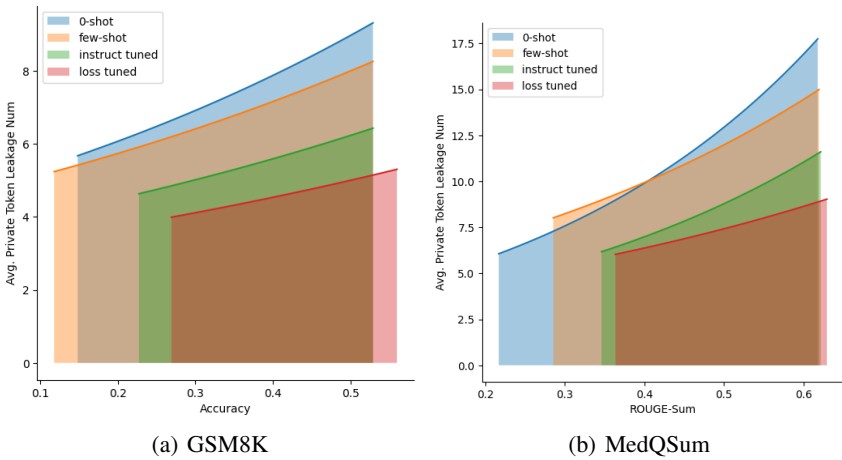

(a) GSM8K        (b) MedQSum

Figure 4: The curves illustrating the relationship between the number of privacy tokens leaked and performance are shown for (a) GSM8K and (b) MedQSum.

| Dataset | Metric | Prompt Engineering | | Instruction Tuning | Loss Tuning |
|---------|--------|--------|--------|--------|--------|
| | | 0-shot | few-shot | | |
| GSM8K | precision | 0 | 64.17 | 82.95 | **91.79** |
| | recall | 0 | 44.20 | 72.89 | **87.24** |
| MedQSum | precision | 48.06 | 68.85 | 85.62 | **90.10** |
| | recall | 8.44 | 42.99 | 68.84 | **82.99** |

Table 3: Privacy identification by different models.

One of the key contributions of our study is the incorporation of multi-objective optimal considerations (e.g., privacy) into the LLM cascade, distinguishing our work from previous approaches. In this section, we demonstrate how these multi-objective considerations help mitigate privacy concerns within the LLM cascade while preserving its ability to enhance performance.

As can be seen in Figure 4, by incorporating privacy considerations into the cascade, the local LLM tends to route a greater proportion of safe queries to the server, as evidenced by the smaller area under the curves for few-shot prompting compared to the area for zero-shot prompting, even when only

a few examples are provided. However, the number of privacy tokens leaked increases at a faster rate compared to the training methods, indicating that relying on the local LLM's in-context ability to identify multiple objectives in cascading is not trustworthy. The privacy identification results presented in Table 3 further validate this claim, as the precision and recall metrics for identifying privacy concerns in queries using prompting methods are not comparable to those of training-based methods. Interestingly, the local LLM $\Phi(L)$ (Gemma-2B) does not recognize personal information, such as names or account details, as privacy concerns, even when explicitly prompted. This oversight could pose risks when the local LLM is applied in real-world financial applications (specific cases can be found in Appendix B). In contrast, the trained $\Phi(L)$ shows significant improvement in identifying private queries, as indicated in Table 3. The gradual increase in performance, illustrated in Figure 4, suggests that the trained $\Phi(L)$ is less likely to route private queries to the server, reinforcing the importance and necessity of incorporating privacy considerations into cascading.

### 4.3 LOGITS DISTRIBUTION STUDY

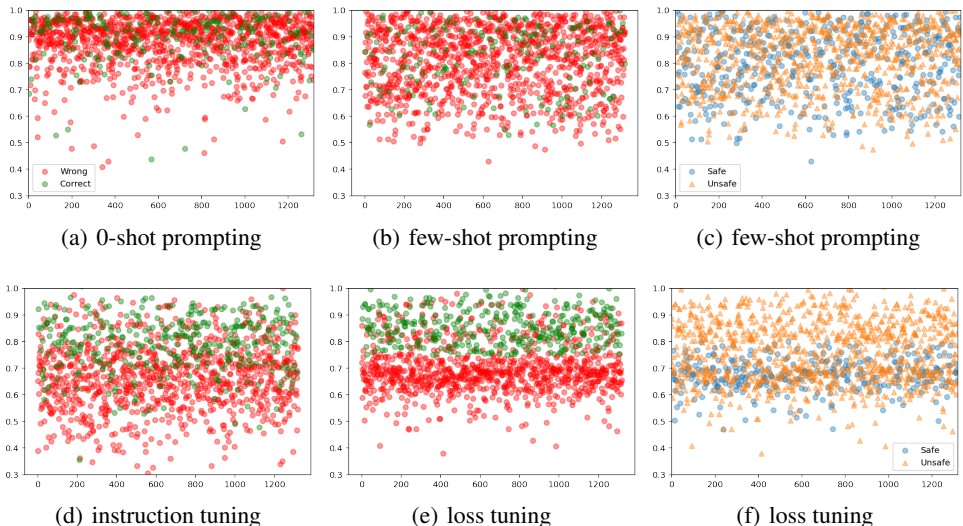

(a) 0-shot prompting      (b) few-shot prompting      (c) few-shot prompting

(d) instruction tuning      (e) loss tuning      (f) loss tuning

Figure 5: Logits scatter distribution produced by different methods on GSM8K dataset. (e) and (f) are logits for privacy concerns; y-axis is the logits, x-axis is the data index.

To further understand the effectiveness of our proposed LLM cascade with multi-objective considerations, we visualize the logit distributions for both training and training-free methods. As shown in Figure 5 and 8, the logits become more decentralized when a few examples are provided for $\Phi(L)$ to learn the cascade logic, in contrast to 0-shot prompting. Additionally, the signals within the distributions for prompting methods are not distinctly separable, which accounts for the randomness observed in routing queries, as discussed in previous sections. In contrast, training methods demonstrate more distinct distributions, where concentrated red points represent the reflection points noted in Figure 3. This indicates that training-based methods better grasp the cascade logic; answers with higher logits are correlated with more correct responses, suggesting that the trained $\Phi(L)$ is more confident in its correct answers and more likely to route difficult queries to the server. Furthermore, the trained model tends to send fewer unsafe queries to the server, as the logits for unsafe responses are generally higher, making them less likely to be sent. These observations reaffirm the effectiveness and necessity of incorporating multi-objective optimal considerations into cascading, highlighting the superiority of our proposed loss function for training the local LLM compared to existing prompting and instruction tuning methods.

## 5 CONCLUSION & FURTURE WORK

In this study, we advance the LLM cascade by incorporating multi-objective optimization, moving beyond existing approaches that primarily emphasize cost-performance trade-offs. This enhance-

ment aligns more closely with the demands of real-world applications. We utilize three methods to assess the necessity and effectiveness of embedding multiple objectives into the cascade. Extensive experiments demonstrate that training is essential for local LLMs to grasp the intricate cascade logic while maintaining their cascading capabilities.

While this work represents the first effort to introduce multi-objective considerations into LLM cascades, future research will explore how the number and complexity of objectives influence the cascade performance of local LLMs. We also aim to develop more sophisticated techniques for integrating these objectives and investigate memory-based methods to sustain favorable cost-performance trade-offs while accommodating a wider array of objectives.

# 6  RELATED WORK

**LLM Cascade** Cascading has been extensively studied and applied across various domains due to its ability to enhance system performance in downstream tasks by selecting appropriate models (Hu et al., 2023; Li et al., 2019; Karlos et al., 2016; Viola & Jones, 2001). Recently, this approach has garnered increasing attention for improving the performance of large language models (LLMs). For instance, Agrawal et al. (2024); Xu et al. (2023); Chen et al. (2024) have explored speculative decoding, which leverages a larger and more powerful LLM to verify token-level accuracy during the inference of a smaller LLM, thereby accelerating the overall process. Despite the success of cascading, researchers have observed that larger, more capable LLMs (e.g., GPT-4 (Achiam et al., 2023)) can be expensive, while smaller LLMs (e.g., GPT-2 (Radford et al., 2019)) may not always meet performance requirements. This has led to the emergence of the deferral rule—determining when to invoke the larger LLM—as a critical area of exploration for balancing performance and cost in LLM cascading (Shekhar et al., 2024; Chen et al., 2023a;b). There are two primary approaches to deferral: confidence-based methods and router-based methods. Confidence-based methods leverage the LLM's confidence in its generated answers to inform deferral decisions. Ideally, an LLM exhibits higher confidence for higher-quality answers, and vice versa. A straightforward approach involves asking the LLM to provide a confidence score alongside its answers, invoking the stronger LLM when the score is low (Zhu et al., 2024). Another prevalent method utilizes the logits of generated tokens to represent the LLM's confidence, with recent studies exploring operations on logits, such as mean (Gupta et al., 2024) and quantile (Jitkrittum et al., 2024). Wang et al. (2024) extended this concept by incorporating the logits of the stronger LLM into the loss function for tuning the weaker LLM, enhancing its understanding of the cascade logic and enabling deferral decisions based on logit indicators. In contrast, router-based methods use a routing mechanism to determine whether to invoke the stronger LLM. Typically, the router selects the most suitable LLM for different tasks. Non-predictive routing evaluates the outputs of multiple LLMs to select the best one, but this can be costly due to the need to assess all involved models (Madaan et al., 2023; Lee et al., 2023; Wang et al., 2023). Predictive routing systems, however, employ reward functions that allow the router to anticipate which LLM to select, thus avoiding the latency associated with extensive evaluations (Shnitzer et al., 2023; Šakota et al., 2024; Hari & Thomson, 2023). Nonetheless, router-based methods require prior knowledge of each LLM's capabilities and may incur significant costs when trying to enhance performance, compared to confidence-based methods (Hu et al., 2024b;a). In this study, we adopt confidence-based methods for LLM cascading.

**Privacy-preservation** Privacy has always been a core concern in LLM research (Kim et al., 2024; Zhang et al., 2024b; Das et al., 2024; Janryd & Johansson, 2024; Feng et al., 2024), particularly for on-device LLM applications (Zhang et al., 2024a; Peng et al., 2024; Yuan et al., 2024). LLMs have been shown to inadvertently reveal sensitive information, such as personal names (Evertz et al., 2024; Kim et al., 2024). To address these privacy issues, Liu et al. (2024a;b;c); Kassem et al. (2023) proposed machine unlearning techniques that enable LLMs to forget sensitive information, thus mitigating the risk of generating harmful or biased content. Another approach is differential privacy, which adds noise to the training data, making it more difficult to identify individual data points (Hartmann et al., 2024). Additionally, Zhang et al. (2024c) suggested using contrastive learning to erase an LLM's memory of user information. While these methods have shown success across diverse user bases, our objective is to enhance the sensitivity of our LLM cascade framework to privacy concerns in single-user settings. To achieve this, we aim to leverage in-context learning and integrate binary privacy identification into the loss function, allowing the local LLM to be more attuned to privacy considerations during the cascading process.

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

## A  PROMPTS

The design of prompts plays a crucial role in activating the LLM's capabilities for downstream tasks. Following the findings of Webson & Pavlick (2021) on prompt design, we first assume a persona for the LLM, then provide task instructions and ask the model to generate outputs in a fixed style. For few-shot prompting, we provide task examples along with their corresponding outputs; details are shown in Fig. 6. Interestingly, we observed that as the number and complexity of tasks in the instructions increased, the model's performance on the target task declined, as demonstrated in Table 2. The prompts presented here yielded the best performance among all the variations we tested.

## B  PRELIMINARY RESULTS

| Metric % | Cascade | Prompt Engineering | | | | Instruction Tuning |
| --- | --- | --- | --- | --- | --- | --- |
| | | 0-shot | 1-shot | 2-shot | 5-shot | |
| Call Rate | | 0 | 70.43 | 48.98 | 67.43 | 42.76 |
| Safe Call Rate | | 0 | 2.05 | 2.94 | 2.13 | **27.61** |
| Accuracy | ✗ | 14.94 | 10.08 | **11.83** | 10.68 | **26.08** |
| | ✓ | 14.94 | 42.91 | 37.30 | 42.61 | 42.29 |

Table 4: Preliminary results on GSM8K.

Following the approach of Hartmann et al. (2024), we initially attempted to use self-critique and rely on the in-context learning capabilities of the local LLM to implement the deferral function. Specifically, we instructed the model to handle the task while simultaneously outputting a confidence level, which would determine whether the query should be deferred to the server. However, preliminary results revealed limitations in this design. As shown in Table 4, without examples, the local model

```
gsm8k_prompt = r'''Assume you're a student working on some mathematical
problems. Now, you'll be giving mathematical problems, you need to do two
tasks: a. Check if the question contains personal information (e.g., names
etc.), output Yes or No only;\n
b. Solve this question; \n

Here are some examples:
Question: Hector purchased a container of gumballs. He gave 4 to Todd,
then he gave twice as many as he had given Todd to Alisha, and then he gave
5 less than four times as many to Bobby as he had given to Alisha. If
Hector had 6 gumballs remaining, what is the total number of gumballs that
Hector purchased? \n
Output:
a. Contains Personal Information: Yes
b. Answer: Hector gave to Alisha twice as many as he had given Todd, for a
total of 4*2=<<4*2=8>>8 gumballs, Hector gave 5 less than four times as
many to Bobby as he had given to Alisha, or a total of
(8*4)-5=<<8*4-5=27>>27 gumballs. If Hector had 6 gumballs remaining, he
originally purchased 4+8+27+6=<<4+8+27+6=45>>45 gumballs. #### 45

Question: A garden produced 237 potatoes, 60 fewer cucumbers and twice as
many peppers than the cucumbers. How many vegetables did the garden
produce? \n
Output:
a. Contains Personal Information: No
b. Answer: 237 potatoes + 60 cucumbers + 2*60 peppers = 237 + 60 + 120 =
317 vegetables.<eos>.

Case,
Question: {question}\n
Output:'''
```

(a) GSM8K Prompt

```
meqsum_prompt = r'''
Please summarize the below consumer health question (CHQ) following the
given examples, output the summarized question only:\n

Examples:
CHQ: SUBJECT: who and where to get cetirizine - D
MESSAGE: I need/want to know who manufscturs Cetirizine. My Walmart is
looking for a new supply and are not getting the recent
Summary: Who manufactures cetirizine?

CHQ: who makes bromocriptine
i am wondering what company makes the drug bromocriptine, i need it for a
mass i have on my pituitary gland and the cost just keeps raising. i
cannot ever buy a full prescription because of the price and i was told if
i get a hold of the maker of the drug sometimes they offer coupons or
something to help me afford the medicine. if i buy 10 pills in which i
have to take 2 times a day it costs me 78.00. and that is how i have to buy
them.  thanks.
Summary: Who manufactures bromocriptine?

Case:
CHQ: {chq}\n
Summary:
'''
```

(b) MedQSum Prompt

```
instruction_prompt_wmt = r'''
Assume you're a professional translator, now please translate the
following sentence into English and check if the sentence contains privacy
leakage. Output the translated sentence and privacy check only.\n
Source sentence: {src}\n
Target sentence:
Privacy:
'''
```

(c) WMT22 Prompting

Figure 6: Prompts Used for Prompt Engineering and Instruction Tuning on three datasets.

tends to be overly confident in every generated response. Moreover, even when provided with several examples, the model treats confidence as a classification task, rather than correlating it with the quality of its generated responses. Consequently, we opted to use logits for more effective LLM cascading. Further, as indicated in section A, as the number and the complexity of tasks within the instruction increase, the model tend to have worse performance on the downstream task. As such, we propose to decompose the tasks within the instruction to several tasks and use different heads to handle it for achieving LLM cascade.

# C SUPPLEMENTARY RESULTS

## C.1 SUPPLEMENTARY CASCADE RESULTS

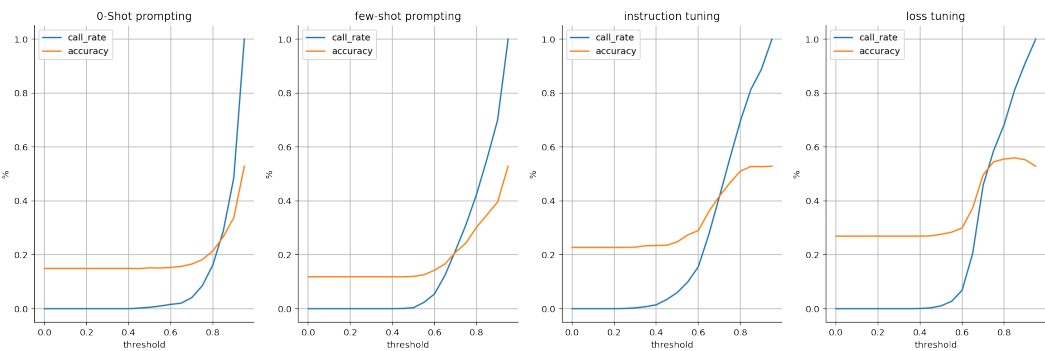

Figure 7: The curve of performance and call rate vs threshold on GSM8K dataset

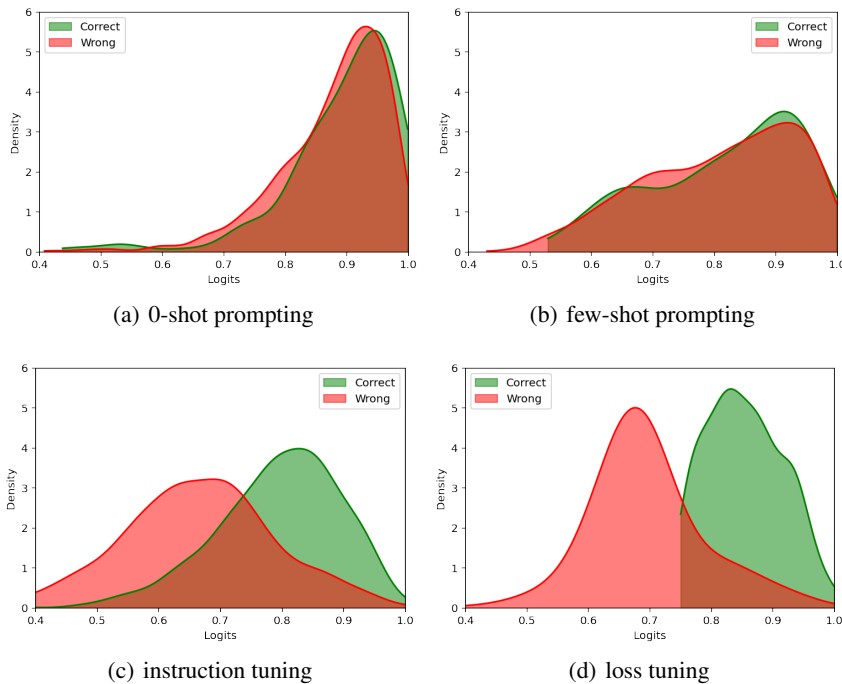

Figure 8: Logits distribution curve by different methods on GSM8K dataset: (a) 0-shot prompting, (b) few-shot prompting, (c) instruction tuning, (d) loss tuning.

As shown in Figure 8, training-based methods have a direct impact on distinguishing between correct and incorrect answers using logits (i.e., the separation between the green and red areas). This aligns with the scatter distribution in Figure 5, further validating the necessity of training in LLM cascading. Additionally, the higher peak in the red area indicates a faster performance improvement, as depicted in Figures 3 and 7. These findings explain the effectiveness and intuition of our approach.

## C.2 DATASETS

| Dataset | Task Type | Avg. Input Length | Avg. Output Length | Avg. Leakage Tokens |
|---------|-----------|-------------------|--------------------|--------------------|
| GSM8K | Question Answering | 52.56 | 83.60 | 5.19 |
| MedQSum | Summarization | 70.51 | 11.49 | 11.27 |
| WMT22 | Translation | 101.67 | 95.19 | - |

Table 5: Statistics of datasets.

Table 5 provides detailed statistics for all datasets. Following the privacy research by Hartmann et al. (2024), we extracted tokens with privacy concerns (e.g., names and other personal identifiers), as the number of such privacy-leakage tokens is critical for evaluating our methods. The extraction was based on PII rules (Kim et al., 2024) and HIPAA regulations (Lincke, 2024), achieving extraction accuracies of 99.1% for GSM8K and 99.7% for MedQSum. A subset of 100 samples was manually verified by a highly educated PhD student, and the p-value score between human and machine extractions was less than 0.05, further validating the effectiveness of our proposed methods.

