# OpenReview forum: "LLM Cascade with Multi-Objective Optimal Consideration"
_ICLR.cc/2025/Conference — Submitted to ICLR 2025_

### Official Review · Reviewer_rmGZ · 2024-10-31

**Soundness:** 2
**Presentation:** 2
**Contribution:** 2
**Rating:** 3
**Confidence:** 5

**Summary:**

The authors of this paper studied the LLM routing problem, where a large and a small LLM are deployed, and the goal is to route the input query to the large LLM only when needed. The practical value of this routing problem is to balance cost and quality, while the authors added another criterion, privacy, into this paper.

In general, the studied problem should be relevant and useful in practice, but the proposed solution does not exhibit necessary technical novelty and the reported experiment results are not supportive. See the detailed comments below.

**Strengths:**

The paper provides relatively rich discussions regarding the related work, and the description of the studied problem is easy to comprehend.

**Weaknesses:**

The idea is rather straightforward: in addition to generating the response, the small model is also trained to predict the outcome of the considered aspects, e.g., whether the query is privacy sensitive, or whether the response is of good quality. Then the deferral module uses the prediction logits to decide if the query should be routed to the large model.

To certain degree, the title is a bit misleading: what the proposed solution addresses is not a multi-objective optimization problem per say, since the routing is actually rule based. And the training of the small LLM is also not addressing a multi-objective optimization problem, but a multi-task learning problem.

The experiment results also look a bit weak: the call rate is very high, especially for the prompt-based and instruction tuning based solutions. And even for the best loss tuning method, more than 80% call rate seems do not really save the cost in practice. Moreover, if I understood correctly, when the query is privacy sensitive, even if the small model’s performance could be bad, we should still not send the query to the large model. 100% call rate in prompt and instruction tuning based methods seem a complete failure? Besides, all the reported the results are from the authors’ proposed methods, shall we also compare some baselines, like those discussed in the introduction and related work?

**Questions:**

How should we set the weights in Eq(1)? And how does the setting of them affect the performance in the reported experiments?

In Figure 3, it seems accuracy is not topped when call rate is 100%. Why is that? In this case it seems very hard to decide when the large or the small LLM would provide the better accuracy?

---

> ### Author Response · Authors · 2024-11-21
> **Thanks for reviewing our work**
>
> Dear Reviewer rmGZ,
>
> Many thanks for your efforts and time on reviewing our work, we sincerely appreciate the insightful feedback and we’re committed to make clarifications here and make improvements in the revision based on your suggestions.
>
> We would like to highlight that the key scenario of our work is in the LLM casacde, of which rare work has been done to consider the deferral decision beyond cost-performance. This leads to our first attempt to include other objectives into the deferral decision making. Our intuitive here is to explore a frame for including more objectives (e.g., privacy). We would like to include more objectives into this frame in the near future, I hope this can better help you understand our work.
>
> **In Responding to Questions**
>
> 1. The weights selection is an experimental process.
>
> 2. Thanks for this question! It by nature won't be the best performance for the call rate is 100%. First, call rate 100% means that all queries are routed to the server LLM, of which the performance will be the server LLM's performance. However, the local LLM and server LLM may have different performance for queries, that is say there exists queries that can be answered correctly by the local LLM while the server LLM is wrong and vice versa. The cascade aims to enable the local llm to route more queries that it can't respond (or has a low confidence in response's quality) and when such queries are routed to the server and answered correctly will lead to a better performance than soley use server llm or local llm. That's also the logic behind the curve of figure 3. The key contribution of our work is that we reduce the privacy token leakage while maintaining the LLM cascade performance.
>
> Hope this better helps you understand our work. Again, we appreciate your feedbacks and will include the discussions in the revision for improvments!

---

### Official Review · Reviewer_2Ekd · 2024-11-04

**Soundness:** 2
**Presentation:** 1
**Contribution:** 2
**Rating:** 3
**Confidence:** 4

**Summary:**

In this submission, the authors propose a multi-objective method to consider more real-world requirements for LLM deployment.

**Strengths:**

1) It is great that the authors consider more real-world requirements for LLM deployment, such as privacy, beyond just performance-cost trade-off. The proposed method has great potentials.

2) Authors design and conduct experiments from several aspect, analyzing the introduced privacy factor.

**Weaknesses:**

1) The writing of this submission can be carefully double-checked and be largely improved.
For example, in Figure 1, the caption describe the opposite contents with subfigures, making readers confusing. And it would be better to have clear-segmented paragraphs, and each one discusses one specific thing, instead of having many different things in a single yet very long paragraph (for example, on the page 2).
When I read the submission, I always feel that sentences close to each other lacks logic, causing difficulty in understanding the submission.

2) Latency should be discussed, and if possible, experimentally tested.
Latency is very important for LLM deployment. It is great that authors realize more requirements should be included, and latency should be one of these requirements. Could the authors please discuss how to incorporate it into the proposed multi-objective optimization? In this way, it will be more convincing as the proposed method considers more than one factor (i.e., only privacy).

3) Experiments can be more real.
The experiments are well designed. However, they are conducted on pseudo scenarios. It will become more solid if the authors can give some experiments for real cases as the motivation of this work is for real-world deployment.

**Questions:**

Please see above Weaknesses.

---

> ### Author Response · Authors · 2024-11-21
> **Thanks for reviewing our work**
>
> Dear Reviewer 2Ekd,
>
> Thank you for your time and efforts on reviewing our work! We sincerely appreciate your constructive feedback and suggestions. We will include the suggestions into consideration accordingly for imrpvoments!

---

### Official Review · Reviewer_R4ss · 2024-11-04

**Soundness:** 3
**Presentation:** 2
**Contribution:** 3
**Rating:** 5
**Confidence:** 4

**Summary:**

This paper presents a multi-objective LLM cascade framework. Traditional LLM cascade systems generally employ smaller, local models for initial processing, only escalating to larger server models if necessary, primarily based on performance-cost considerations. Here, the authors propose a novel approach that integrates additional objectives, such as privacy, into cascade decision-making.
The methodology involves training and training-free techniques, including prompt engineering, instruction tuning, and a custom loss-tuning mechanism that aligns with multi-objective goals. Experimental results, conducted on benchmarks like GSM8K and MedQSum, show that training-based methods outperform prompt engineering, achieving higher accuracy and safe call rates while controlling privacy leakage.

**Strengths:**

- The key strength of this work is its approach to incorporating multi-objective considerations, including privacy, into the LLM cascading framework.
- The paper explores a mix of training-based and training-free techniques—like prompting, instruction tuning, and loss tuning. Practitioners can choose the methods that best fit their computational resources and deployment constraints.

**Weaknesses:**

* This paper builds on the work of Wang et al. (2024), which introduced cascade-aware training, i.e., a technique where the smaller model is fine-tuned with an understanding of its role within the cascade and the capabilities of the larger supporting model, achieved through knowledge distillation. Although Wang et al. (2024) is cited, a more thorough comparison between the two approaches would strengthen the analysis. Based on my understanding, this paper extends Wang et al.'s cascade-aware framework by incorporating privacy as an additional weighted component in the loss function for local training, i.e., the first summation term in Eqn (1) in the paper.

* The technical writing could be improved for clarity. I only understood this paper’s loss-tuning approach after reading Wang et al. (2024). Clearly positioning this paper within the context of existing work would greatly aid readers in understanding its contributions.

* Appropriately assigning weights to different objectives and determining the threshold in the multi-objective cascade system is very difficult, thus, limiting its usefulness in practice.

    *  As more objectives are added, it becomes nearly impossible to interpret trade-offs. Moving from a curve in two dimensions (performance vs cost) to higher-dimensional trade-off surfaces complicates understanding.
    * It will be helpful to include a 3D surface plot or a contour plot that visualizes trade-offs between performance, cost, and privacy, with contour lines showing constant privacy values across the performance-cost plane. This helps illustrate how adjustments in performance and cost impact privacy.
    * The paper does not provide a concrete, practical method for tuning weights and thresholds to achieve the desired balance between objectives. The absence of guidelines means that applying this framework in practice can be a trial-and-error process, which is time-intensive and resource-consuming.

Reference
- Congchao Wang, Sean Augenstein, Keith Rush, Wittawat Jitkrittum, Harikrishna Narasimhan, Ankit Singh Rawat, Aditya Krishna Menon, and Alec Go. Cascade-aware training of language models. arXiv preprint arXiv:2406.00060, 2024

**Questions:**

See above

---

> ### Author Response · Authors · 2024-11-21
> **Thanks for reviewing our work**
>
> Dear Reviewer R4ss,
>
> Thank you for your time and efforts on reviewing our work! We sincerely appreciate your constructive feedback and suggestions. We will include our discussions into the revision accordingly for imrpvoments.
>
> Thanks for acknowledging the contributions of our work in the scenario of LLM cascade, we're committed to improve the writing in the revision. We want to highlight that the loss design for multi-objective tuning for LLM cascade reduce the privacy leakage greatly while maintaining the cascade performance as can be seen in the Section 4.1 and Section 4.2. Further, the Figure 3 and Figure 5 witness the curve of maintaining LLM cascade performance. Beyond the LLM cascade performance, we alse observed the private tokens leakage mitigation which further validate the success of our explorations in including more objectives into deferral decision making. We will include the 3D figure into the revision for better presentation and hope this helps you better understand the contribution of works.
>
> Thank you!

---

### Official Review · Reviewer_ad5R · 2024-11-08

**Soundness:** 1
**Presentation:** 1
**Contribution:** 1
**Rating:** 3
**Confidence:** 3

**Summary:**

The paper aims to improve the existing cascading based LLM inference by adding objectives beyond accuracy. The paper starts off by making the point that existing cascading literature for the most part focuses on accuracy and uncertainty while in real world, many more objectives like privacy might be relevant. The paper proposes a simple multi objective loss function where different objectives are weighed using different coefficients. The paper shows experiments with three real world datasets showing how the privacy and task performance are balanced.

**Strengths:**

1. Privacy-aware inference is an important problem.
2. The paper provides extensive background on LLM cascades and quite a bit of detail on the related work.

**Weaknesses:**

While the paper tackles an interesting problem, it has three fundamental issues which mean it is not quite ready for publication yet.

First, the writing and presentation of the paper needs to be massively improved. The paper provides too much information when it is not needed, and worse, too little information where it is actually needed. For instance, the intro spends a lot of time on details which are irrelevant to the core contribution of the paper, e.g., the discussion of model confidence in line 71. However, in line 185, the introduction of distillation loss is very abrupt and comes without context. It is not clear to the readers how the models are being distilled, e.g., what data is being used for distillation? Up until this section, distillation was not really mentioned. This results in a situation where the reader cannot grasp the core setup of the models, e.g., training and fine-tuning details.

Second, from what this reviewer could understand, the only contribution of the paper is to turn the regular accuracy oriented (cascade-based) training into multi-objective training. Multi-objective training of ML models (e.g., the type shown in Eq. 1) is not new and has been considered in areas like fairness beforehand, see for instance, [here](https://dl.acm.org/doi/10.1145/3278721.3278779) and [here](https://arxiv.org/abs/2106.12639). So the technical novelty of the paper does not seem sufficient for ICLR.

Third, the paper could do a better job at motivating the setup considered here. It does describe what precisely the privacy concerns in the datasets considered here are. What kind of personal information does the GSM8K dataset contain? How sensitive is this information? Who are the stakeholders and what are their concerns? What is the privacy threat model? Why is the token-based leakage a good solution? The paper provides the reader to Appendix C2 for more details, which contains no examples showing privacy leakage and creates more questions than answers. For instance, what was the p-value computed on?

**Questions:**

1. Line 189: On “optionally” adding the distillation loss, when is it actually added?

2. Eq 1: When applying the method proposed here on a new dataset, how should one select the weight?

3. Line 141: Why does each of the objective generate a new response? Shouldn’t all objectives be covered in a single response?

---

> ### Author Response · Authors · 2024-11-21
> **Thank you for reviewing our work**
>
> Dear Reviewer ad5R,
>
> Many thanks for your efforts and time on reviewing our work, we sincerely appreciate the insightful feedback and we’re committed to make clarifications here and make improvements in the revision based on your suggestions.
>
> First of all, we would like to highlight that the key scenario of our work is in the LLM casacde, of which rare work has been done to consider the deferral decision beyond cost-performance. This leads to our first attempt to include other objectives into the deferral decision making. Our intuitive here is to explore a frame for including more objectives (e.g., privacy). We have made a detailed background elaboration in the Introduction Section, assisting readers to better understand the scenarios of our work and our contributions.
>
> The dataset is detailed in Section 3.1, GSM8K contains privacy concerns such as names etc, MedQSum contains privacy concerns such as name/personal medical records etc. Our method reduce the leaksge of such information when the local LLM is producing response to users.
>
> **In Responding to the Questions**
>
> 1. As detailed in line 200-202, the loss is controled by a factor \alpha, that is say when the logit (i.e. confidence) of generated response is not high then the distillation loss will be added.
>
> 2. The loss weight selection is an experimental process
>
> 3. The objectives are related to the losses, the LLM only generate a single response. For instruction tuning and prompting, we ask the LLM to produce responses with a fixed style and for the loss tuning, we use different heads to endorse such considerations to LLM when do training. The generated response for all methods are single.
>
> Again, we sincerely appreciate your insightful feedsbacks and will include our discussions into the revision accordingly. Hope this response addresses your concerns!

---

### Meta-Review · Area_Chair_gXza · 2024-12-20

**Metareview:**

This paper aims to improve the cascading in LLM deployment. While prior solution focuses on balancing cost and quality of the model’s outputs by designing routing techniques between large and small LLMs, this paper introduces the privacy into the considered criteria.

Reviewers all concerned about the marginal contribution by incorporating privacy losses in prior frameworks, which I agree. But I think the more important issues are the presentation of the paper, which can improve significantly by restructuring some local information, and the presented experiments, which need better design to show practical relevance.

**Additional Comments On Reviewer Discussion:**

Most of the reviewers’ concerns remain after the rebuttal and discussions.

---

### Decision · Program_Chairs · 2025-01-22

Reject